# Modeling Event-level Causal Representation for Video Classification

Yuqing Wang
yuqing_wang@mail.sdu.edu.cn
Shandong University
Jinan, China

Lei Meng*
lmeng@sdu.edu.cn
[1]Shandong University
[2]Shandong Research Institute of
Industrial Technology
Jinan, China

Haokai Ma
mahaokai@mail.sdu.edu.cn
Shandong University
Jinan, China

Yuqing Wang
wang_yuqing@mail.sdu.edu.cn
Shandong University
Jinan, China

Haibei Huang
huanghaibei@inspur.com
Inspur
Jinan, China

Xiangxu Meng
mxx@sdu.edu.cn
Shandong University
Jinan, China

## Abstract

Classifying videos differs from that of images in the need to capture the information on what has happened, instead of what is in the frames. Conventional methods typically follow the data-driven approach, which uses transformer-based attention models to extract and aggregate the features of video frames as the representation of the entire video. However, this approach tends to extract the object information of frames and may face difficulties in classifying the classes talking about events, such as "fixing bicycle". To address this issue, This paper presents an Event-level Causal Representation Learning (ECRL) model for the spatio-temporal modeling of both the in-frame object interactions and their cross-frame temporal correlations. Specifically, ECRL first employs a Frame-to-Video Causal Modeling (F2VCM) module, which simultaneously builds the in-frame causal graph with the background and foreground information and models their cross-frame correlations to construct a video-level causal graph. Subsequently, a Causality-aware Event-level Representation Inference (CERI) module is introduced to eliminate the spurious correlations in contexts and objects via the back- and front-door interventions, respectively. The former involves visual context de-biasing to filter out background confounders, while the latter employs global-local causal attention to capture event-level visual information. Experimental results on two benchmarking datasets verified that ECRL may better capture the cross-frame correlations to describe videos in event-level features. The source codes have been released at https://github.com/wyqcrystal/ECRL.

## CCS Concepts

• **Computing methodologies → Artificial intelligence**.

---

*Corresponding author.

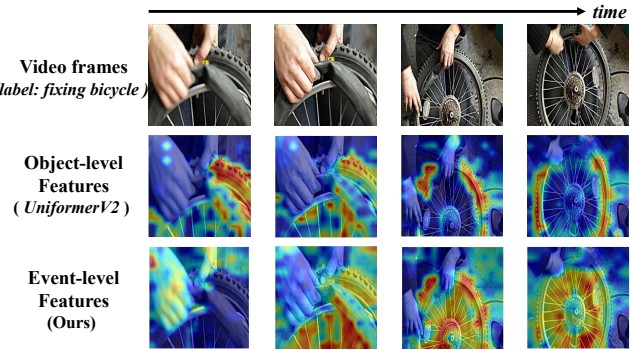

**Figure 1: Object-level features v.s. Event-level features. In the case of "fixing bicycle" from ActivityNet, UniformerV2 misclassifies it as "assembling bicycle". It stems from overly focusing on the object-level feature of the "whole wheel".**

## Keywords

Video Classification, Causal Intervention, Event-level Modeling

**ACM Reference Format:**

Yuqing Wang, Lei Meng, Haokai Ma, Yuqing Wang, Haibei Huang, and Xiangxu Meng. 2024. Modeling Event-level Causal Representation for Video Classification. In *Proceedings of the 32nd ACM International Conference on Multimedia (MM '24), October 28-November 1, 2024, Melbourne, VIC, Australia.* ACM, New York, NY, USA, 9 pages. https://doi.org/10.1145/3664647.3681547

## 1 Introduction

Video classification aims to automatically identify and categorize video content into predefined categories. This enables the efficient retrieval of video from the extensive corpus of the internet. Videos are not just a collection of static images, but a comprehensive concept characterized by the cross-frame interconnections among scenes, events, and objects. In video classification, the modeling of any specific entity is invariably affected by the presence of the other two types of entity, thereby impeding its process of representation learning. Therefore, how to learn the discriminate representations of the predicted targets from the complex spatio-temporal dynamics is a permanent challenge in video classification tasks.

To address the spatio-temporal dynamics issue in video classification, existing studies attempt to achieve the video understanding by aggregating all the frame representations via the spatio-temporal

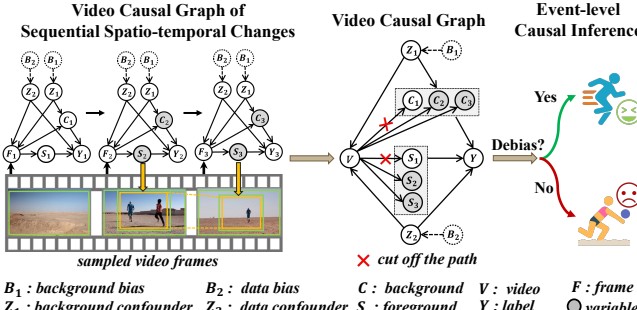

**Video Causal Graph of Sequential Spatio-temporal Changes**

**Video Causal Graph**

**Event-level Causal Inference**

*sampled video frames*

$B_1$ : background bias   $B_2$ : data bias   $C$ : background   $V$ : video   $F$ : frame
$Z_1$ : background confounder   $Z_2$ : data confounder   $S$ : foreground   $Y$ : label   ◯ variable
× *cut off the path*

**Figure 2: Illustration of the proposed ECRL. The inherent background bias and data bias from video sequences may lead to incorrect prediction, like "playing volleyball". In contrast, ECRL employs the multi-perspective causal interventions at the event level to effectively cut off the paths $V \rightarrow S_1$ and $V \rightarrow C_1$, thereby enabling precise prediction as "running".**

convolution [11, 28] and the self-attention mechanism [2, 54, 64]. The first type of these studies conducts the sparse sampling on video sequences to achieve the local feature aggregation [43]. The second type fully leverages the unsaturated discriminative power of residual attention to understand the long-term temporal dependencies in video sequences[17]. As stated in the second line of Fig. 1, these aforementioned methods are profoundly influenced by the configurations of their visual encoders, which results in an excessive concentration on the object knowledge within each frame. This leads to a mechanical aggregation of object-level features from each frame, thereby impeding their in-depth exploration of cross-frame event correlations. In contrast, the event-level representation in the third line of Fig. 1 can precisely highlight the position of "hand" and "whole wheel" during the video sequences modeling, thereby correctly understanding the category of "fixing bicycle".

Causal representation learning (CRL), benefiting from its interpretability, has achieved outstanding performance in video understanding, including Video Moment Retrieval [61] and Video Question Answering [26, 27]. This arises from the capacity of CRL to disentangle video representations in a frame-by-frame manner through the logical reasoning process. It effectively eliminates the inherent confounding factors that precipitate errors in model recognition, thereby facilitating the mining of event-level knowledge from the temporal correlations among objects within the video sequences. These methods typically focus on leveraging the additional textual knowledge to construct the Structural Causal Model (SCM), thereby guiding the visual representation modeling by the multi-model representation alignment [25]. However, these above CRL-based methods struggle to be directly applied in video classification tasks where semantic information is unavailable. Therefore, how to utilize CRL to model the event-level knowledge from the video sequences remains an urgent problem to be addressed.

To address this issue, we propose a **E**vent-Level **C**ausal **R**epresentation **L**earning (ECRL) model for video classification, which leverages the causal theory to discover the association among inter-frame objects in video sequences and guide the event-level representations modeling in current video classification algorithms. As illustrated in Fig. 2, ECRL is comprised of two principal components: the Frame-to-Video Causal Modeling (F2VCM) module and

the Causality-aware Event-level Representation Inference (CERI) module. Specifically, F2VCM first systematically extracts the background and foreground content from the raw video sequences to highlight the indispensable knowledge in each frame. To discover the event-level causal correlation, it simultaneously models the in-frame object interactions and the cross-frame object correlations on these continuous back- and fore-ground frames to construct an event-level causal graph. Recognizing the spurious associations between the ground-truth category with the background elements and the intra-frame knowledge, we argue that video classification is subject to background biases and data biases in video sequences. Therefore, CERI is introduced to alleviate these inherent biases via the Background Debiasing (BD) component and the Global-Local Causal Attention (GLCA) component. Firstly, BD conducts backdoor interventions to reduce the spurious associations such as "beach, sky -> playing volleyball", thereby alleviating the background biases. In contrast, GLCA implements front-door interventions to remove data biases within the inter-frame object correlations, highlighting the indispensable frames and filter the dispensable frames. With F2VCM and CERI, ECRL is able to effectively constructs the video-level causal graph with the intra-frame information and inter-frame correlations, thereby enabling the structured interventions to model the informative event-level causal representations.

We have conducted extensive experiments on two benchmark datasets to verify the effectiveness of ECRL, including performance comparison, ablation study, in-depth analyses, and case studies. These discoveries demonstrate the efficacious mechanism of ECRL, which models video-level causal structures with in-frame object interactions and cross-frame correlations. Additionally, it refines event-level causal representations through multi-perspective causal interventions, thereby enhancing the accuracy of video classification tasks. To summarize, this paper includes three contributions:

- We propose an Event-level Causal Representation Learning method to model the event-level spatio-temporal representations from the causal perspective. To the best of our knowledge, this is the pioneering solution that models the event-level representation via causal theory in video classification.

- We introduce two effective F2VCM and CERI modules to construct the video-level causal graph with the intra- and inter-frame correlations and model the event-level causal representation via the multi-perspective causal interventions.

- We conduct the visualization analysis to highlight the ability of ECRL to mitigate the erroneous predictions stemming from the inherent background bias and data bias, which enables the precise event-level video understanding of ECRL.

## 2 Related works

### 2.1 Video Classifiaction

Video classification aims to find category-related information from successive frames. With the development of deep learning[13, 14, 29–32, 34, 60], convolutional neural networks have made great breakthroughs in the field of video classification. TSN [47]suggests utilizing a sparse sampling strategy for capturing long-range video segments. I3D [8]advances 2D CNN models by extending them to 3D CNNs. Non-local [50] employs a non-local block to capture long-range dependencies in the video through the non-local mean

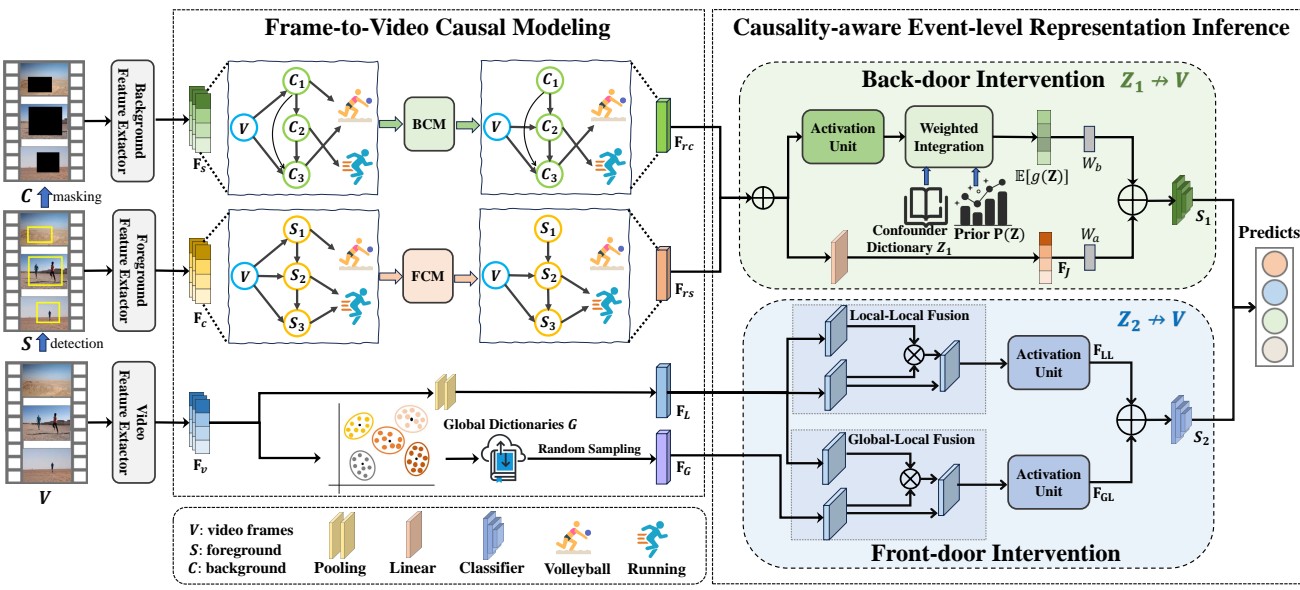

**Figure 3: Illustration of the framework of ECRL. ECRL proposes a Frame-to-Video Causal Modeling (F2VCM) module to build the video-level causal graph with the in-frame object interactions and the cross-frame object correlations. To alleviate the inherent background and data bias in video sequences, ECRL proposes a Causality-aware Event-level Representation Inference (CERI) module to eliminate these spurious correlations via the back-door and the front-door interventions.**

operation. S3D [57]introduces temporal separable convolution and spatio-temporal feature gating to improve the performance of I3D. TSM [21] introduces temporal shift operations. SlowFast [12] consists of a slow path and a fast path. TEA [19] adjusts spatio-temporal features with motion features to enhance motion patterns. X3D [11] expands 2D networks across dimensions including time, space, depth, and width. The emergence of Transformers has advanced video classification. ViViT [2] introduces a transformer model for video classification, offering variants that consider both temporal and spatial dimensions. TimeSformer [5] uses individual attention mechanisms within each patch block, aiding in modeling long sequence videos. Swin Transformer [27] merges CNNs and transformers, converting spatio-temporal attention using localized induction bias and non-overlapping windows.

Existing methods explore spatio-temporal video dynamics but overlook video and background biases, limiting event-related representation and performance. Thus, learning causal event representations via cross-frame correlations is crucial for video classification.

## 2.2 Causal Inference in Video Understanding

Causal inference is an analytical tool designed to infer the dynamics of events under changing conditions (e.g. different treatments or external interventions) [36]. Compared to conventional debiasing techniques[49, 53], causal inference[37, 51, 61] shows its potential in reducing spurious correlations[4] and disentangling model effects[6], thereby achieving better generalisation performance. Causal inference and counterfactual reasoning have received increasing attention in a variety of tasks in computer vision[18, 22, 38], including visual explanation[20, 24, 25, 39, 52], scene graph generation[55], image recognition [9, 18, 33, 40, 65, 66],

video analysis, and visual-language tasks[1, 62]. Existing causal theories are relatively mature in the field of images and are also widely applied in the domain of videos. Causal graphs are used to establish causal-effect inference for multimodal video summarization [16]. Dataset selection bias can result in spurious correlations between speech and video in video grounding tasks [35]. Causal inference is used to address imbalances in the distribution of the datasets. The counterfactual-based model ensemble in video anomaly detection integrates long-term and local image contexts for effective anomaly detection [56]. Causal methods in video moment detection are used to address spurious correlations between long-tail annotations, user queries, and moment positions.[61]. Causal methods are widely used in Video Question Answering (VQA) [55] due to biases in datasets related to visual, linguistic, and annotation aspects. These methods enhance the robustness of visual question answering by exploring potential causal relationships in complex spatio-temporal scenarios.

However, existing causal methods in the video domain focus on finding correlations between visual and textual data, require information other than the video data (e.g., textual information ), and thus cannot be directly applied to video classification tasks. The challenge of video classification is the representation of events. It is important to design a causal model that is suitable for the video classification task and to find the directly related to the event information representation via the causal mechanism.

## 3 Problem formulation

For the video classification task, given a dataset $\mathcal{D} = \{\mathcal{V}_i \mid i = 1, \ldots, N\}$, labels $\mathcal{Y} = \{y_i \mid i = 1, \ldots, J\}$, and $\mathcal{V} = \{v_i \mid i = 1, \ldots, t\}$. Conventional methods extract visual features

from the video: $\mathbf{F}_v = \mathcal{M}_v(\mathcal{V})$, where $\mathcal{M}_v(\cdot)$ denotes the feature extractor. Then, the category mapping $\mathcal{P}(\cdot)$ predicts the category of the sample, i.e. $\mathcal{P}(\mathbf{F}_v) \rightarrow \bar{y}$.

Different from the conventional approaches, the proposed video classification method ECRL, based on event-level causal representation learning, first utilizes obtaining the foreground and background information $\mathcal{V}_s = \{s_i \mid i = 1, \ldots, t\}$, and $\mathcal{V}_c = \{c_i \mid i = 1, \ldots, t\}$. Then, the original frame-level of visual features $\mathbf{F}_v = \mathcal{M}_v(\mathcal{V})$, foreground features $\mathbf{F}_s = \mathcal{M}_s(\mathcal{V}_s)$ and background feature $\mathbf{F}_c = \mathcal{M}_c(\mathcal{V}_c)$, where $\mathcal{M}_s(\cdot)$ and $\mathcal{M}_c(\cdot)$ denote the visual feature extraction network. To construct the causal graph of the video, we further model the relation between $\mathbf{F}_s$ and $\mathbf{F}_{rs} = \mathcal{R}_s(\mathbf{F}_s)$, $\mathbf{F}_{rc} = \mathcal{R}_c(\mathbf{F}_c)$ to get content-level representations of $\mathbf{F}_{rs}$ and background $\mathbf{F}_{rc}$, where, $\mathcal{R}_s(\cdot)$ and $\mathcal{R}_c(\cdot)$ denote the networks for cross-frame correlation modeling. To address background bias and data bias in videos, we process $\mathbf{F}_{rs}$ and $\mathbf{F}_{rc}$ via BD module and GLCA module, obtain the background debiased feature $\mathbf{F}_{cc}$ and the causal representation $\mathbf{F}_{gg}$. Finally, we calculate the predictive classification results by mapping the two features $\mathcal{P}(\mathbf{F}_{cc}) + \mathcal{P}(\mathbf{F}_{gg}) \rightarrow \bar{y}$.

## 4 Methods

ECRL introduces an Event-level Causal Representation Learning method to enhance the model's causal awareness of event information. The architecture of ECRL is illustrated in Fig. 3. An event-level causal graph is constructed using the Frame-to-Video Causal Modeling (F2VCM) module, which explores the event correlations by finding the correlation between the foreground and background of the video frames; the Causality-aware Event-level Representation Inference (CERI) module eliminates background and data bias in video data by implementing causal intervention on the causal graph. This enables the model to find information directly related to event representation and the causal structure in video classification tasks.

### 4.1 Frame-to-Video Causal Modeling Module

The Frame-to-Video Causal Modeling (F2VCM) module aims to build the in-frame causal graph and learn cross-frame causal correlations to create a Structural Causal Model (SCM) suitable for video classification to assist causal inference. It consists of two main processes: extracting related visual elements and constructing content-level representations by modeling inter-frame correlations between the foreground and background visual elements.

*4.1.1 Visual Representation Learning.* This module aims to extract three types of visual information: foreground, background, and original video frame features. Given a data set of videos $\mathcal{D} = \{\mathcal{V}_i \mid i = 1, \ldots, N\}$, we distinguish the foreground $\mathcal{S}_i$ from $\mathcal{V}_i$ using a pre-trained Saliency model [23] and then implement a masking process to obtain the corresponding contextual background $C_i$, where $\mathcal{V}_i$ is a video sample. Next, we extract features using the specified encoder.

$$F_{vi} = \mathcal{M}_v(\mathcal{V}_i), F_{si} = \mathcal{M}_s(\mathcal{S}_i), F_{ci} = \mathcal{M}_c(C_i) \quad (1)$$

where $\mathcal{M}_v(\cdot)$, $\mathcal{M}_s(\cdot)$, and $\mathcal{M}_c(\cdot)$ represent the encoders for the original video, foreground, and background frames respectively. $F_{vi}, F_{si}, F_{ci} \in \mathbb{R}^{t \times d}$, t denotes the number of frames, and $d$ denotes the dimension of the extracted features.

*4.1.2 Cross-frame correlation modeling (CCM) module.* We model causal graph element correlations to extract content-level representations for event-level causal inference. The CCM employs separate BCM (Background Cross-frame Modeling) and FCM (Foreground Cross-frame Modeling) modules for different tasks.

In this section, we use * to refer to foreground S and background C. For the video frame information $F_{vi}$, the video content features are obtained by averaging $F_L = \delta(F_{vi})$ with the dimension $F_L \in \mathbb{R}^d$. Then we adopt a context generator $\mathcal{T}_*(\cdot)$ to obtain the context vector and input it together with $F_{*i}$ into the attention frame weighting module to obtain the weight $w_*$. Next, $w_*$ is used to weight $F_{*i}$, resulting in the foreground representation of the entire video sequence, obtained through association modeling:

$$\begin{aligned} w_* &= \phi\left(\mathcal{H}\left(\mathcal{T}_*\left(F_{*i}\right), F_{*i}\right)\right) \\ F_{r*} &= \rho\left(w_* \odot F_{*i}, \delta\left(F_{*i}\right)\right) \end{aligned} \quad (2)$$

where $\mathcal{H}(.,.)$ denotes the Hadamard inner product operation, $\phi$ denotes the softmax function, $\odot$ represents the dot product operation and $\rho$ means concat and linear operation. We obtain the representations for video, foreground, and background $F_L, F_{rs}, F_{rc} \in \mathbb{R}^d$.

*4.1.3 Causal View at Video Classification Task.* We use the structural causal model to model the variable relationships of complex spatiotemporal data in video classification tasks. As illustrated in Fig. 4 (a), it is a directed acyclic graph $\mathcal{G} = \{\mathcal{N}, \mathcal{E}\}$ in which the nodes $\mathcal{N}$ denote variables and edges $\mathcal{E}$ denote association between variables. The SCM $\mathcal{G}$ includes six variables: video $V$, foreground and background feature $S$, $C$, background and data confounder $Z_1$, $Z_2$, and prediction $Y$. The correlations in graph $\mathcal{G}$ are as follows:

$B_1 \rightarrow Z_1 \rightarrow V$, $B_2 \rightarrow Z_2 \rightarrow V$: $B_1$ and $B_2$ represent the background and data bias, leading to spurious correlations between videos and categories. To cut off the correlation between $V$ and $\{Z_1, Z_2\}$, the causal intervention is necessary (Fig. 4 (b) and (c)).

$Z_1 \rightarrow C \rightarrow Y$: $C$ represents the background confounder. $Z_1 \rightarrow C$ means harmful confounding factors making unreliable context learning, further influencing the prediction of $Y$ via the link $C \rightarrow Y$.

$V \rightarrow C \rightarrow Y, V \rightarrow S \rightarrow Y$: $S$ represents the foreground feature. These two causal pathways reflect the estimation of $Y$ based on unbiased contextual features $C$ and content of video frames $V$.

$V \leftarrow Z_1 \rightarrow C \rightarrow Y$: We use the backdoor intervention to cut off the adverse backdoor path and eliminate the impact of $Z_1$, which misleads the spurious "context->category" correlation (Fig. 4 (b)).

$M \leftarrow V \leftarrow Z_2 \rightarrow Y$: We use a mediator M to construct the front door path and adopt front door intervention to mitigate the spatio-temporal data bias in the video. (Fig. 4 (c)).

Training a video classification model focuses on learning true causal effects. It aims to discern category $Y$ based on key events in the video, avoiding spurious correlations from confounders $Z_1$ and $Z_2$. By cutting off these two paths, the causal structure in video classification is discovered (Fig. 4 (d)).

### 4.2 Causality-aware Event-level Representation Inference Module

Through the implementation of the F2VCM module, we have established an event-level SCM and conducted targeted causal interventions to derive precise event causal representations. The

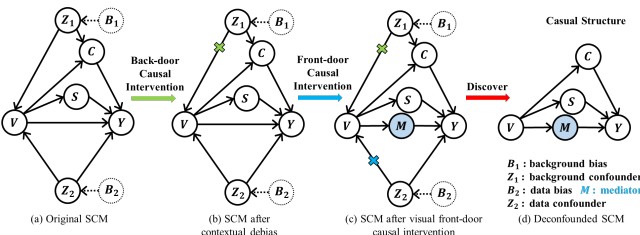

**Figure 4: The event-causal intervention process of CERI module for video classification.**

Causality-aware Event-level Representation Inference (CERI) module needs to model the event-level representation of the video. First, we employ a backdoor intervention through a Background Debiasing (BD) module to eliminate background biases. Then, we applied a front-door intervention using the Global-Local Causal Attention module (GLCA) to remove data biases. Ultimately, we discovered the SCM for representation learning in video classification, which enhances the model's ability to model event-level information.

*4.2.1 Background Debiasing (BD) Module by Back-door Causal Intervention.* To alleviate the background bias We propose a Visual Context Debiasing (BD) module, which can remove the side effects caused by the background confounder and facilitate a fair contribution of diverse backgrounds to video understanding. Implementation details are described below.

In Fig. 4 (b), existing video classification methods use the conditional probability $P(Y \mid V)$, expressed by Bayes' rule.

$$P(Y \mid V) = \sum_z P(Y \mid V, S = f_{rs}(V), C = f_{rc}(V, z)) P(z \mid V) \quad (3)$$

where $f_{rs}(\cdot)$ and $f_{rc}(\cdot)$ represent CCM module encoding functions. To mitigate this observation bias caused by $Z_1$, an intuitive idea is to intervene on $V$ to ensure that each contextual semantics contributes equally to the video classification. To address this issue, we stratify $Z_1$ based on backdoor adjustments[37], and then estimate the average causal effect based on the context proportion of training samples and implement intervention on $V$:

$$P(Y \mid do(V)) = \sum_z P(Y \mid V, S = f_{rs}(S), C = f_{rc}(V, z)) P(Z_1) \quad (4)$$

where $do(V)$ denote intervene on $V$. By implementing do-calculus on $V$, the path in Fig. 4 (b) from $Z_1$ to $V$ is cut-off, and the model will approximate causal intervention $P(Y \mid do(V))$, rather than a spurious association $P(Y \mid V)$.

**Confounder Dictionary $Z_1$.** Given the difficulty of capturing all real-world confounders, we use a stratified confounder dictionary $Z_1 = [z_1, z_2, \ldots, z_L]$ structured as an $L \times d$ matrix, where $L$ is the context category size and $d$ is the background feature dimension, pre-trained using the UniformerV2 model. For practical applications, we define the confounder pool as $M_b = \{m_k \mid k = 1, \ldots, N\}$, with $N$ representing the number of training samples in the dataset. We apply k-means++[3] and principal component analysis [41] to derive $Z_1$ from the confounder pool, ensuring each $z_i$ reflects contextual clustering. Specifically, each $z_i$ is the average feature of its cluster, calculated as $z_i = \frac{1}{\sigma_i} \sum_{j=1}^{\sigma_i} m_j^i$, where $\sigma$ is the count of contextual features in the $i$-th cluster.

**Instantiation of the Proposed BDM.** To implement the theoretical and imaginative interventions in Eq 4. The BDM is described as follows. As the computation of $P(Y \mid do(V))$ requires multiple pass backs for all z, to reduce computation, we use Normalised Weighted Geometric Mean (NWGM) [59] to approximate the results expected from the above feature layers:

$$P(Y \mid do(V)) \overset{\text{NWGM}}{\approx} P(Y \mid V, S = f_{rs}(V), C = \sum_z f_{rc}(V, z) P(z)) \quad (5)$$

We parameterize the network model to approximate the conditional probability of Eq. 5, inspired by[48], as follows:

$$P(Y \mid do(V)) = W_a J + W_b \mathbb{E}_z[g_1(z)] \quad (6)$$

where $W_a \in \mathbb{R}^{d_m \times d_a}$ and $W_b \in \mathbb{R}^{d_m \times d}$ are learnable parameters. Additionally, $J = \psi(s, c) \in \mathbb{R}^{d_a \times 1}$, where $\psi(\cdot)$ represents a fusion strategy, integrating $s$ and $c$ into the joint representation $J$. The above approximation is justified as the influence on $Y$ comes from $v$, $c$, and the confounding factor $z$. Therefore, we approximate $\mathbb{E}_z[g_1(z)]$ as a weighted integration of all background prototypes.

$$\mathbb{E}_z[g_1(z)] = \sum_{i=1}^N \mu_i z_i P(z_i) \quad (7)$$

where $\mu_i$ represents the important weight coefficient measuring the interaction between each $z_i$ and the original feature $J$, with $P(z_i) = N_i / N_m$. In practice, we employ two implementation approaches for $\mu_i$ : Dot Product (softmax $\left[ (W_c J)^T (W_d z_i) / \sqrt{d} \right]$) and Additive Attention (softmax $\left[ W_e^T \cdot \text{Tanh}(W_c J + W_d z_i) \right]$), where $W_e \in \mathbb{R}^{d_n \times 1}$, $W_c \in \mathbb{R}^{d_n \times d_a}$, and $W_d \in \mathbb{R}^{d_n \times d}$ are mapping matrices.

By using the contextual debiasing module, we end up with a debiased visual feature of $F_{cc}$. We then input this feature into the classifier to get the predicted score $S_1$, where $S_1 = \text{classifier}(F_{cc})$.

*4.2.2 Global Local Causal Attention(GLCA) Module by Front-door Causal Intervention.* It is very complex and difficult to observe this data bias $Z_2$, where not all content in the video is assigned to a category. Fortunately, the front-door adjustment is a feasible way to compute $P(Y \mid V)$ when $Z_2$ is unobservable (Fig. 4 (c)). Insert a mediator M between V and Y, we can construct a front door path $V \rightarrow M \rightarrow Y$ to transmit the knowledge, where $m$ denotes the knowledge selected from M:

$$P(Y \mid V) = \sum_m P(M = m \mid V) P(Y \mid M = m) \quad (8)$$

The classification predictor can be represented by $V \rightarrow M$ and $M \rightarrow Y$. The intervention probability $P(Y \mid do(V))$ as follows:

$$P(Y \mid do(V)) = \sum_m P(M = m \mid do(V)) P(Y \mid do(M = m)) \quad (9)$$

In Fig. 4(c), for the causal link $V \rightarrow M$, the backdoor path between V and M is blocked by $M \leftarrow V \leftarrow Z_2 \rightarrow Y$. Therefore, the intervention probability is equal to the conditional probability.

$$P(M = m \mid do(V)) = P(M = m \mid V) \quad (10)$$

$M \rightarrow Y$: To achieve $P(Y \mid do(M = m))$, we can indirectly cut off the link $M \leftarrow V$ to block the backdoor path $M \leftarrow V \leftarrow Z_2 \rightarrow Y$.

$$P(Y \mid do(M = m)) = \sum_v P(V = v) P(Y \mid V = v, M = m) \quad (11)$$

In summary, by applying Eq.9 and Eq.10 into Eq.11, we can compute the true causal effect between V and Y.

$$P(Y \mid \mathrm{do}(M = m)) = \sum_m P(M = m \mid V)$$
$$\times \sum_v P(V = v)P(Y \mid V = v, M = m) \quad (12)$$

We parameterize $P(Y \mid V, M)$ as $g_2(\cdot)$ followed by a softmax layer, as most visual tasks are formulated similarly.

$$P(Y \mid V, M) = \phi[g_2(M, V)] \quad (13)$$

where $\phi$ denotes the softmax function. To address the cost of forwarding for all samples is prohibitively high, we still adopt the NWGM approximation, as shown in the formula.

$$P(Y \mid do(V)) \approx \varphi[g_2(\hat{M}, \hat{V})]$$
$$= \varphi\left[g_2\left(\sum_m P(M = m \mid f(V))m, \sum_v P(V = v \mid h(V))v\right)\right] \quad (14)$$

where $\hat{M}$ and $\hat{V}$ stand for the estimates of $M$ and $V$, while $f(\cdot)$ and $h(\cdot)$ denote the network mapping functions.

**Global Clustering Dictionary** $G$: To improve representation of causally perceived visual features, the GLCA module estimates M and V within a unified attention framework. Local features $F_L$ come from the input, while global features $F_G$ are sampled from a K-means initialized and training-updated dictionary $G$. M and V are derived from local-local $F_{LL}$ and local-global $F_{LG}$ features, respectively. We illustrate using $F_{LG}$.

First, the global-local fusion obtains the fusion feature $h = [W_v F_G, W_q F_L \odot W_k F_G]$, where $[.,.]$ denotes the join operation, $\odot$ denotes the Hadamard inner product, $W_q, W_k, W_v, W_{h'}$ denote the linear layer weights. Then it goes to the attention unit attention weights $\alpha$, and weights the local features to obtain the global-local feature $F_{LG}$, the process is as follows:

$$F_{LG} = \phi\left(RELU\left(W_h h + b_h\right)W_{h'} + b_{h'}\right) \odot F_G \quad (15)$$

where $b_h$ and $b_{h'}$ denote the linear layer bias. The local-local visual feature $F_{LL}$ can be obtained and combined with $F_{LL}$ and $F_{LG}$ to obtain the causal representation after the visual front door intervention as $F_{gg}$. We then input this feature into the classifier to get the predicted score $S2$, where $S_2 = \text{classifier}(F_{gg})$.

The bias $B_1$ and $B_2$ in the video are eliminated using the Event-detected Causal Graph Construction Module. This results in the causal-aware structure for video classification shown in Fig. 4(d).

### 4.3 Training Strategies

In this module, the main objective is to integrate the predicted scores $S_1$, $S_2$, obtained from the BD and GLCA modules respectively, to derive the final video classification category. we fusion the scores $S_1, S_2$, obtaining the $S$, where $S = S_1 + S_2$. The prediction $p = \frac{e^S}{\sum_j e^S}$, where $p$ represents the predicted final outcome. A single-stage training model is used to update the model by minimizing the prediction loss $\mathcal{L}_{ce} = CE(p, p')$, where CE is the cross-entropy loss, $p$ is the predicted result, and $p'$ is the ground truth label.

Training ECRL from scratch is challenging due to uneven feature distributions. To address this, we use a two-step strategy: pre-train on raw video frames to extract features and calculate losses, then

**Table 1: Statistics of the two wide-used datasets.**

| Datasets | #Class | #Training | #Testing |
|----------|--------|-----------|----------|
| MSR-VTT | 20 | 7,010 | 2,990 |
| ActivityNet | 100,000 | 10,009 | 4,515 |

fine-tune the causal intervention framework, enhancing stability and effectiveness.

## 5 Experiments

### 5.1 Experiment Settings

*5.1.1 Datasets.* We conduct extensive experiments on video classification datasets MSR-VTT [58] and ActivityNet[7] to verify the effectiveness of ECRL. Their statistics are shown in Table 1. MSR-VTT dataset comprises 10,000 video clips. We split it into 9,000 and 1,000 videos for training and testing. ActivityNet consists of 20,000 videos. We clipped the video durations to 0-30 seconds. We remove video samples that lack labels. We split the data into 1,009 and 4,515 videos for training and testing, respectively.

*5.1.2 Evaluation Criteria.* Following conventional video classification methods [64], we leverage the Accuracy@k with $k = \{1, 5\}$ to evaluate the video classification performance.

*5.1.3 Implementation Details.* Following the pre-training of the large model of UniformerV2 [17], we set 768 as the feature dimension. We use the Adam optimizer with a learning rate chosen from 1e-6 to 1e-4. The decay rate of the learning rate parameter is selected from 0.1 and 0.5, and the decay interval is 4 epochs. The batch size is set to 8, and the UniformerV2-based model requires 4-6 hours for training. For confounder dictionary construction, the backdoor confounder dictionary is sized at 256 for MSR-VTT and 512 for ActivityNet, while the fron-tdoor dictionary $Z_1$ is uniformly set at 512 with a dimension of 768 across all datasets.

### 5.2 Performance Comparison

We compare ECRL with 12 video classification methods, including X3D[11], TANet[28], TDN[46], GC-TDN[15], H$^2$CN [44], LAPS[63], ViT[10], ViViT[2], TokShift[64], TimeSformer[5], Video-Swin[27], and UniformerV2[17]. Note that UniformerV2 is one of the SOTA methods, and leverage it as the base model. Only the ViT model leverages pre-trained ImageNet weights, while the remaining models utilize weights pre-trained on Kinetics400. Specifically, for ViViT and X3D, we sample 16 video frames for input. For the other models, we select 8 video frames as input. The best performance is marked in bold. The following observations can be drawn from Table 2:

- ViT-based architectures outperform ResNet50-based ones on two datasets. This is due to Vision Transformers employing a self-attention mechanism that captures global dependencies in videos, free from the constraints of fixed convolutional structures.
- UniformerV2 outperforms ViT-based architectures by addressing the local video redundancy issues inherent in Vision Transformers (ViTs), using a combination of convolution and self-attention within a transformer framework.

**Table 2: Performance comparison of baselines and ECRL .**

| Method | backbone | MSR-VTT | | ActivityNet | |
|---|---|---|---|---|---|
| | | ACC@1 | ACC@5 | ACC@1 | ACC@5 |
| X3D | X2D | 51.17 | 81.67 | 71.71 | 92.00 |
| TANet | ResNet50 | 53.47 | 81.20 | 76.13 | 94.06 |
| TDN | ResNet50 | 50.94 | 81.00 | 73.08 | 91.84 |
| GC-TDN | ResNet50 | 52.17 | 82.14 | 75.42 | 93.79 |
| $H^2CN$ | ResNet50 | 51.97 | 81.67 | 73.51 | 92.92 |
| Video-Swin | Swin-B | 56.42 | 84.75 | 83.80 | 96.37 |
| LAPS | ViT-B | 54.11 | 82.58 | 79.51 | 95.58 |
| ViT | ViT-B | 54.54 | 81.20 | 80.12 | 95.18 |
| ViViT | ViT-B | 55.79 | 84.95 | 80.96 | 95.46 |
| TokShift | ViT-B | 56.56 | 84.35 | 82.03 | 95.93 |
| TimeSformer | ViT-B | 55.92 | 82.54 | 82.76 | 96.05 |
| UniformerV2 | ViT-B | 60.73 | 85.75 | 86.77 | 96.94 |
| ECRL | ViT-B | **63.41** | **86.72** | **89.01** | **97.68** |

- ECRL demonstrates superior performance on two distinct datasets due to its meticulous modeling of category-related event-level information in video classification, enhancing the model's causal perception and overall efficacy.
- The ECRL achieves its most notable performance enhancement on the MSR-VTT. This improvement is primarily due to the dataset's wide variety of events, highlighting our model's ability to accurately capture event information in videos.

## 5.3 Ablation Study

In this section, we further studied the working mechanism of each module of ECRL, as shown in Table 3. The best performance is marked in bold. The following findings could be observed:

- Using only foreground (+FCM) or background information (+BCM) significantly reduces model performance, highlighting that background information is crucial for complete video representation. However, this reliance can also cause confusion.
- Associating fine-grained visual frames (+FCM+BCM+TSC) is crucial for integrating principal and background frames (+FCM+BCM). Temporal modeling reduces background-label correlations, improving essential visual data capture.
- Background debiasing (+FCM+BCM+BD) mitigates bias in video classification. Modeling relationships across frames (+FCM+BCM+TSC+BD) enhances the BD module's effectiveness, allowing more accurate measurement of true causal effects.
- ECRL captures event-level causal representations using SCM by constructing causal graphs, modeling frame relationships, and employing causal interventions to eliminate biases.

## 5.4 In-depth Analyses

### 5.4.1 Effectiveness Analyses of the background confounders dictionary $Z_1$.
As delineated in Table 5, we scrutinized the efficacy of the background confounder dictionary $Z_1$ within the BD module. Experimental results show that replacing $Z_1$ with a random dictionary significantly deteriorates performance, validating our contextual prototypes. Using average video features as a confounder dictionary is less effective than class-average background features, indicating that random dictionaries and class averages are insufficient context confounders and highlight the impact of inherent background biases on model generalization.

**Table 3: Ablation study of ECRL on UniformerV2. ECRL is equal to "+ FCM + BCM + TSC + BD + GLCA". FCM: foreground only, without cross-frame modeling. BCM: background only, without cross-frame modeling; TSC: cross-frame modeling with background and foreground; BD: Background Debiasing Module; GLCA: Global-Local Causal Attention Module.**

| model | MSR-VTT | | ActivityNet | |
|---|---|---|---|---|
| | ACC@1 | ACC@5 | ACC@1 | ACC@5 |
| base | 60.73 | 85.75 | 86.77 | 96.94 |
| + FCM | 58.90 | 85.61 | 85.10 | 96.66 |
| + BCM | 56.38 | 83.04 | 80.34 | 94.58 |
| + FCM + BCM | 60.33 | 85.82 | 86.69 | 96.47 |
| + FCM + BCM + TSC | 60.60 | 85.93 | 86.80 | 96.51 |
| + FCM + BCM + BD | 61.24 | 85.42 | 87.63 | 97.21 |
| + FCM + BCM + TSC + BD | 62.94 | 86.36 | 88.55 | 97.60 |
| ECRL | **63.41** | **86.72** | **89.01** | **97.68** |

**Table 4: The result of different components in $\mathbb{E}_z[g_1(z)]$.**

| Setting | MSR-VTT | | ActivityNet | |
|---|---|---|---|---|
| | ACC@1 | ACC@5 | ACC@1 | ACC@5 |
| w/o $\mu_i$ | 62.00 | 85.82 | 87.43 | 97.46 |
| w/o P $(z_i)$ | 62.51 | 86.26 | 87.59 | 97.50 |
| w/Additive Attention | 62.71 | 86.29 | 87.73 | 97.52 |
| BD | **62.94** | **86.36** | **87.92** | **97.62** |

**Table 5: The results on different versions of confounder dictionary $Z_1$ in BD. Random Dictionary, Average Dictionary, and Background Dictionary use random features, average visual features, and the average features of the background class as confounders, respectively.**

| Setting | MSR-VTT | | ActivityNet | |
|---|---|---|---|---|
| | ACC@1 | ACC@5 | ACC@1 | ACC@5 |
| Random Dictionary | 57.93 | 86.25 | 82.76 | 96.05 |
| Average Dictionary | 60.00 | 85.62 | 86.02 | 97.23 |
| Background Dictionary | **62.94** | **86.36** | **87.92** | **97.62** |

### 5.4.2 Effectiveness Analyses of the $\mathbb{E}_z[g_1(z)]$.
To validate the effectiveness of $\mathbb{E}_z[g_1(z)]$ each component in the context prototype integration as shown in Table 4, we conducted experiments by removing weights $\mu_i$ and prior probabilities $P(z_i)$ to assess the effectiveness of the weighted integration. The results indicate that identifying the importance and proportion of each confounder is crucial for effective causal intervention. It was also found that both ad_causal and dp_causal paradigms are meaningful and applicable.

## 5.5 Case Study

### 5.5.1 Quality Analysis of the Causal Inference Representation Learning.
To illustrate the differences between $P(Y|X)$ and $P(Y|do(X))$ after causal inference interventions, we used t-SNE [45] to visualize the feature distribution of specific categories (movie, animation, animals, kids, cooking) in the MSR-VTT test set. As Fig. 5 shows, before causal inference, features for movie and animation categories were mixed due to the diversity of scenes and creative backgrounds, causing confusion with other categories and adding data biases. Using BD and GLCA module, ECRL removed

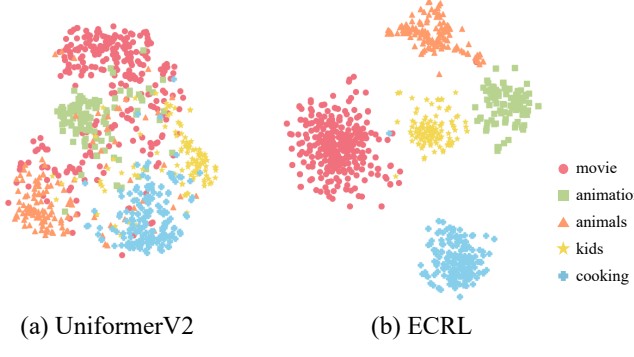

(a) UniformerV2      (b) ECRL

**Figure 5: Visualization results of UniformerV2 and ECRL on the MSR-VTT dataset.**

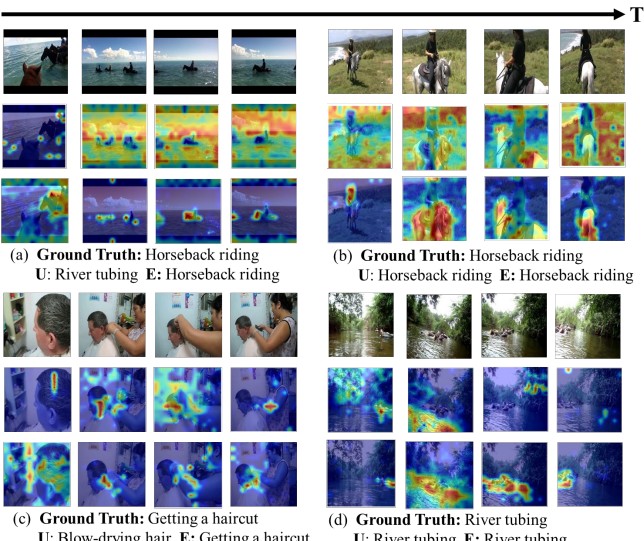

(a) **Ground Truth:** Horseback riding
U: River tubing   **E:** Horseback riding

(b) **Ground Truth:** Horseback riding
U: Horseback riding   **E:** Horseback riding

(c) **Ground Truth:** Getting a haircut
U: Blow-drying hair   **E:** Getting a haircut

(d) **Ground Truth:** River tubing
U: River tubing   **E:** River tubing

**Figure 6: Visualization of attention on sampled frames from the ActivityNet. Three lines respectively represent the original video frame, the corresponding attention map presented by the UniformerV2 (U) and our ECRL (E).**

these biases, making feature distributions distinct and enabling the learning of causal representations of events.

*5.5.2*   ***Visualization of the Event-level Causal Representation by ECRL***. To evaluate ECRL's efficacy, we compared it with UniformerV2 on the ActivityNet validation dataset, examining their attention to causally relevant visual information in videos. Using GradCAM[42] to generate heatmaps, we observed significant differences. ECRL mitigates erroneous predictions caused by background biases, as seen in Fig.6 (a) and (b). The base model incorrectly associates the background (sky and seawater) with "River tubing," which ECRL corrects. ECRL focuses on crucial visual information linked to specific events, reducing biases. In Fig.6 (b) and (d), ECRL captures foreground details better, enhancing event causality representation. Fig.6 (c) shows ECRL's strong causal perception, focusing on verbs and nouns in "Getting a haircut," effectively connecting related event information across frames.

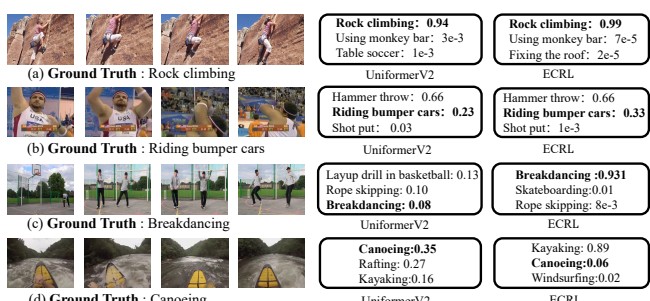

(a) **Ground Truth** : Rock climbing

| UniformerV2 | ECRL |
|---|---|
| Rock climbing: 0.94 | Rock climbing: 0.99 |
| Using monkey bar: 3e-3 | Using monkey bar: 7e-5 |
| Table soccer: 1e-3 | Fixing the roof: 2e-5 |

(b) **Ground Truth** : Riding bumper cars

| UniformerV2 | ECRL |
|---|---|
| Hammer throw: 0.66 | Hammer throw: 0.66 |
| **Riding bumper cars: 0.23** | **Riding bumper cars: 0.33** |
| Shot put: 0.03 | Shot put: 1e-3 |

(c) **Ground Truth** : Breakdancing

| UniformerV2 | ECRL |
|---|---|
| Layup drill in basketball: 0.13 | **Breakdancing :0.931** |
| Rope skipping: 0.10 | Skateboarding:0.01 |
| **Breakdancing: 0.08** | Rope skipping: 8e-3 |

(d) **Ground Truth** : Canoeing

| UniformerV2 | ECRL |
|---|---|
| **Canoeing:0.35** | Kayaking: 0.89 |
| Rafting: 0.27 | **Canoeing:0.06** |
| Kayaking:0.16 | Windsurfing:0.02 |

**Figure 7: Case Study on ECRL in successful and failure cases. To qualitatively demonstrate the effectiveness of ECRL in mitigating background and data bias and discovering event-related information, we select four frames of four video samples from ActivityNet. (a) Both models make correct predictions. (b) Both models make wrong predictions. (c) Only ECRL makes the correct prediction. (d) Only UniformerV2 makes the correct prediction.**

*5.5.3*   ***Error Analysis of ECRL***. We analyzed both accurate and erroneous instances within the ActivityNet dataset. Fig.7 shows that the visualizations illustrate ECRL's robust capacity for spatio-temporal reasoning and its efficacy in diminishing fallacious correlations. In scenarios where the subject is prominent and the backdrop of the video frames remains relatively static (refer to Fig.7 (a)), both the baseline model and ECRL render correct determinations, albeit ECRL with enhanced precision. When the subject is ambiguous (Fig.7(b)), neither model performs well. ECRL effectively neutralizes background biases, correcting the baseline model's misclassification of a basketball court as "Layup drill in basketball" to "breakdancing" (Fig.7(c)). However, in cases of incomplete subjects and static backgrounds (Fig.7(d)), ECRL can still make errors due to over-intervention.

## 6 Conclusion

We propose ECRL (Event-level Causal Representation Learning), which constructs event-level causal graphs through in-frame and cross-frame causal learning. ECRL uses backdoor and front-door interventions to mitigate background and data biases, revealing the causal structure in video classification. This approach eliminates various biases in continuous video sequences, generating informative causal representations. Experimental results confirm ECRL's effectiveness in reducing biases and capturing cross-frame correlations for event-level video features.

In the future, we will also continue to investigate the long-term contextual relationships within complex spatio-temporal environments. Moreover, how to fully leverage the informative multi-modal knowledge within the video sequences is also one of our future research directions.

## Acknowledgments

This work is supported in part by the Shandong Province Excellent Young Scientists Fund Program (Overseas) (Grant no. 2022HWYQ-048), and the Oversea Innovation Team Project of the "20 Regulations for New Universities" funding program of Jinan (Grant no. 2021GXRC073).

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
