# OpenReview forum: "Modeling Event-level Causal Representation for  Video Classification"
_acmmm.org/ACMMM/2024/Conference — MM2024 Oral_

### Official Review · Reviewer_5eRF · 2024-05-18

**Rating:** 4
**Confidence:** 3

**Summary:**

The paper introduces an Event-level Causal Representation Learning (ECRL) model for video classification, addressing the limitations of conventional methods that struggle with event-level classification. ECRL incorporates two main components: the Frame-to-Video Causal Modeling (F2VCM) module and the Causality-aware Event-level Representation Inference (CERI) module. F2VCM builds a video-level causal graph by extracting and modeling foreground and background information. CERI eliminates spurious correlations via backdoor and front-door interventions. Experimental results on benchmark datasets demonstrate that ECRL effectively captures cross-frame correlations and reduces biases, enhancing event-level video classification accuracy.

**Strengths:**

Innovative Approach: The ECRL model introduces a novel way to handle video classification by focusing on event-level causal relationships, which is a significant improvement over traditional object-level methods. This is also the first time I have seen problem-solving from a causal reasoning perspective in video classification tasks, the paper's approach is relatively innovative.

Effective implementation: The combination of F2VCM and CERI modules offers a robust framework that captures both intra-frame and inter-frame correlations, addressing both background and data biases. The paper effectively combines ideas with relevant implementation techniques of deep learning.

Empirical Validation: The experimental results on multiple benchmark datasets (MSR-VTT and ActivityNet) show that ECRL outperforms state-of-the-art methods, validating the effectiveness of the proposed approach.

Detailed Analysis: The paper provides extensive ablation studies, in-depth analyses, and visualizations to explain the contributions of different components of ECRL. These experiments further confirm that the various components proposed by the author are not redundant. For the visualization of the heat map in Figure 6, it is impressive to see the advantages of the proposed framework compared to traditional frameworks.

**Limitations:**

Complexity: The proposed model's complexity, involving multiple modules and interventions, may pose challenges in terms of computational efficiency . Therefore, you should add a comparison term for the number of model parameters or computational complexity in Table 2.

Paper expression: There are certain problems in the writing and image expression of the article.

(1) Firstly, the lack of a period in line 191 is an issue that cannot be ignored.

(2) Although the paper drew many figure, the mismatch between the explanations and the figure actually caused greater reading confusion.

(a). Like line144-line147, the modules involved in this explanation (at least the abbreviations of the modules) should be indicated in Figure.

(b). What are the full names of BCM and FCM in Figure 3? I did not see a clear full name in the article. From the content, should lines 385-397 describe these two modules? In addition, the title CCM module of 4.1.2 is not indicated in Figure 3.

(c). Line 497，What is the full name of BDM?

(d). 4.2.2，Which part of the figure 3 does GLCA correspond to? It should be indicated in the figure.

(3)  For computer vision papers, the formulas in the article are biased towards theoretical descriptions, which is not a conference manuscript in the field of machine learning. Formulas that lean towards mathematical descriptions should not be used.

A far fetched explanation: Why does the model only focus on the hand in the first frame of the visualization in Figure 1?

Note: The readability of a manuscript largely determines its quality! I hope the author can correct the expression issues in the manuscript, I still tend to improve my score.

**Suitability:**

2

---

### Official Review · Reviewer_W4M2 · 2024-05-26

**Rating:** 4
**Confidence:** 1

**Summary:**

This paper proposes to classify actions with causal methods. Specifically, it extracts three representations from foreground (objects), background (scene), and full videos. With these three representations, it builds in-frame and cross-frame causal graphs. Experiments on several video understanding benchmarks show the effectiveness of the methods.

**Strengths:**

1. Unlike current methods that use neural networks for video classification, this paper uses causal graphs for video content analysis, which provides better explainability.
2. Experimental results on several video classification benchmarks show the effectiveness of these methods.

**Limitations:**

1. The causal representation is quite complex for video classification. Does it have limitations in applying to different benchmarks？(robustness)
2. The presentations of method could be further improved, current version is kind of complex and not easy to follow.

**Suitability:**

2

---

### Official Review · Reviewer_Atsm · 2024-06-01

**Rating:** 4
**Confidence:** 3

**Summary:**

The submission introduces a novel model, ECRL, which focuses on event-level causal representation for video classification. The ECRL consists of two modules i.e., F2VCM and CERI. The F2VCM module constructs an in-frame causal graph and models cross-frame correlations to build a video-level causal graph. Another one is the CERI module which uses backdoor interventions for visual context de-biasing and front-door interventions through global-local causal attention to focus on event-level visual information. The classification results on MSR-VTT and ActivityNet verify the effectiveness of ECRL.

**Strengths:**

- The proposed ECRL with its F2VCM and CERI modules, offers a well-structured methodology for discovering causal correlations and reducing biases in video classification tasks.
- The submission is well-structured, with a clear presentation of the problem, methodology, experiments, and results.

**Limitations:**

Cons:
- The difference between CRL-based Video QA and ECRL should be clarified. What are the major differences between CRL-base Video QA and the proposed ECRL? It seems that it is simple to adopt a video CRL-based video QA model for video classification since it also has the ability for video understanding. More specifically, what are the differences between CMCIR [1] and the proposed ECRL, except for the missing text modality?
- The reasons for evaluating the proposed method on MSR-VTT and ActivityNet should be clarified. Do the categories of them contain any event-level information? These two datasets seem not to be used often for video classification.

Typos:
 - Line 115, 'multi-model' -> 'multi-modal'

Ref:
[1] Cross-modal causal relational reasoning for event-level visual question answering.

**Suitability:**

2

---

### Official Review · Reviewer_dxiM · 2024-06-24

**Rating:** 5
**Confidence:** 3

**Summary:**

The manuscript proposes an Event-level Causal Representation Learning (ECRL) model for video classification, which leverages the causal theory to discover the association among inter-frame objects in video sequences and guide the event-level representations modeling. The ECRL manuscript model consists of two principal components: the Frame-to-Video Causal Modeling (F2VCM) module and the Causality-aware Event-level Representation Inference (CERI) module. The F2VCM module builds the in-frame causal graph and learns cross-frame causal correlations to create a Structural Causal Model (SCM) suitable for video classification to assist causal inference. The CERI module eliminates background and data bias in video data by implementing causal intervention on the causal graph. The ECRL model is able to effectively construct the video-level causal graph with the intra-frame information and inter-frame correlations, thereby enabling the structured interventions to model the informative event-level causal representations.

**Strengths:**

The paper is technically sound, presenting a clear methodology for constructing and utilizing causal graphs. The use of back-door and front-door interventions to eliminate spurious correlations showcases a deep understanding of causal inference techniques. The integration of these techniques within a video classification framework is executed with precision, ensuring the model's reliability and correctness.

**Limitations:**

1. The author seems to have not provided an analysis of the computational complexity of the proposed model. The evaluation focuses primarily on accuracy improvements. However, other important metrics such as computational efficiency, memory usage, and latency are not thoroughly discussed. Including these metrics would provide a more comprehensive understanding of the model's performance and practicality.
2. It is necessary to distinguish vectors and scalars in the formula. Please use bold to indicate vectors.
3. The interpretability of the ECRL model is poor, making it difficult to understand the decision-making process of the model. Although causal graphs are inherently more interpretable than purely data-driven approaches, the complexity and high-dimensional nature of video data can still make the resulting graphs difficult to interpret.
4. How to integrate the results of backdoor and front door interventions. What impact will different fusion methods have on the results?

**Suitability:**

2

---

### Meta-Review · Area_Chair_HvZ3 · 2024-07-03

**Recommendation:** Accept (Oral)
**Confidence:** 5

**Metareview:**

This paper proposes a novel approach for video classification that learns causal representations by integrating temporal and contextual cues from multiple modalities, to enhance video understanding and classification.
The paper is well-structured, and clearly written.  It has a high degree of novelty and it is technically sound.
All reviewers recommended to accept the paper and I recommend it for oral presentation.